# Mechanisms of the Device Property Alteration Generated by the Proton Irradiation in GaN-Based MIS-HEMTs Using Extremely Thin Gate Insulator

**DOI:** 10.3390/nano13050898

**Published:** 2023-02-27

**Authors:** Sung-Jae Chang, Dong-Seok Kim, Tae-Woo Kim, Youngho Bae, Hyun-Wook Jung, Il-Gyu Choi, Youn-Sub Noh, Sang-Heung Lee, Seong-Il Kim, Ho-Kyun Ahn, Dong-Min Kang, Jong-Won Lim

**Affiliations:** 1ICT Components & Material Research Laboratory, Photonic/Wireless Convergence Research Department, Electronics and Telecommunications Research Institute, Daejeon 34129, Republic of Korea; 2Korea Multi-Purpose Accelerator Complex, Korea Atomic Energy Research Institute, Gyungju 38180, Republic of Korea; 3Department of Electrical/Electronic, University of Ulsan, Ulsan 44610, Republic of Korea; 4Department of IT Convergence, Uiduk University, Gyeongju 38004, Republic of Korea

**Keywords:** GaN, Si_3_N_4_, HfO_2_, gate insulator, MIS-HEMT, total ionizing dose effects, displacement damages, proton, radiation effects

## Abstract

Recently, we reported that device performance degradation mechanisms, which are generated by the γ-ray irradiation in GaN-based metal-insulator-semiconductor high electron mobility transistors (MIS-HEMTs), use extremely thin gate insulators. When the γ-ray was radiated, the total ionizing dose (TID) effects were generated and the device performance deteriorated. In this work, we investigated the device property alteration and its mechanisms, which were caused by the proton irradiation in GaN-based MIS-HEMTs for the 5 nm-thick Si_3_N_4_ and HfO_2_ gate insulator. The device property, such as threshold voltage, drain current, and transconductance varied by the proton irradiation. When the 5 nm-thick HfO_2_ layer was employed for the gate insulator, the threshold voltage shift was larger than that of the 5 nm-thick Si_3_N_4_ gate insulator, despite the HfO_2_ gate insulator exhibiting better radiation resistance compared to the Si_3_N_4_ gate insulator. On the other hand, the drain current and transconductance degradation were less for the 5 nm-thick HfO_2_ gate insulator. Unlike the γ-ray irradiation, our systematic research included pulse-mode stress measurements and carrier mobility extraction and revealed that the TID and displacement damage (DD) effects were simultaneously generated by the proton irradiation in GaN-based MIS-HEMTs. The degree of the device property alteration was determined by the competition or superposition of the TID and DD effects for the threshold voltage shift and drain current and transconductance deterioration, respectively. The device property alteration was diminished due to the reduction of the linear energy transfer with increasing irradiated proton energy. We also studied the frequency performance degradation that corresponded to the irradiated proton energy in GaN-based MIS-HEMTs using an extremely thin gate insulator.

## 1. Introduction

Gallium nitride (GaN)-based high electron mobility transistors (HEMTs) have been studied for high-power radio frequency (RF), low noise, and aerospace applications, since GaN shows a high breakdown electric field, high carrier density and mobility at the hetero-interface, and a wide bandgap [1,2,3]. In GaN-based HEMTs, many dangling bonds exist at the AlGaN barrier surface, which traps negative charges [4,5]. The negative trapped charges reduce the 2-dimensional electron gas (2DEG) density and degrade the RF performance, which is so-called because of current collapse effects [4,5]. Furthermore, the Schottky contact, which is applied to the gate electrode, limits the drain current driving capacity and output power characteristics, as the gate leakage current is high [6,7,8,9].

In order to overcome these issues, the dielectric layer deposition, which is located on top of the AlGaN barrier and acts as a passivation layer, is essentially required in GaN-based HEMTs. The dielectric layer screens the dangling bonds and reduces the current collapse [4,5], which improves the RF performance in GaN-based HEMTs. In addition, a GaN-based metal–insulator–semiconductor (MIS)-HEMTs structure has been proposed. In GaN-based MIS-HEMTs, the dielectric layer that plays as a gate insulator is inserted between the AlGaN barrier and the gate electrode. The GaN-based MIS-HEMTs show lower gate leakage current, improved drain current driving capacity, and output power performance, which can be compared to the GaN-based HEMTs [10,11]. From this perspective, various gate layers and multi-layered gate insulator structures have been investigated in GaN-based MIS-HEMTs [10,11,12,13,14,15,16,17,18].

However, the usage of the dielectric layer results in a few side effects in GaN-based MIS-HEMTs. The device performance and reliability deteriorate during the device operation since the dielectric layer quality is degraded by the hot-electron-related trapping [19]. In addition, the employment of the dielectric layer should be more cautious, especially for aerospace applications. The radiation resistance of the dielectric layer is weaker than that of semiconductors such as AlGaN and GaN [20,21]. Therefore, various gate dielectric layers, such as SiO_2_ [22], Si_3_N_4_ [23], Al_2_O_3_ [24], Gd_2_O_3_ [25], MgCaO [26], and SiN/Al_2_O_3_ multi-layered gate insulator structure [27], have been studied for aerospace applications.

When GaN-based HEMTs and MIS-HEMTs are exposed to proton irradiation, two different radiation effects are induced. One is the total ionizing dose (TID) effect and another is the displacement damage (DD) effect. The TID effects are mostly induced inside the dielectric layer. The dielectric layer quality is degraded and charges are trapped in the interior of the dielectric layer by the radiation, which deteriorates device performance [20,22,26]. Whereas the DD effects, which are related to the point defects generation, occur in semiconductors [28,29,30], which results in the reduction of the 2DEG density and device performance degradation [23,24,25,27]. When the radiation is irradiated in a GaN-based device, Ga and N vacancies and interstitials are generated and added to the existing point defects. To reveal the device property alteration mechanisms, which are caused by the DD effects, various studies have been conducted [19,21,22,23,24,25,27,31,32,33,34]. However, the device performance alternation mechanisms are not clear in many cases; so far, the TID and DD effects are simultaneously generated and the impact of the two different radiation effects on the GaN-based HEMTs and MIS-HEMTs characteristics are not the same. 

In this study, the device property alteration and its mechanisms, which were generated by proton irradiation, were investigated in GaN-based MIS-HEMTs using an extremely thin gate insulator. Our optimized dielectric layer deposition process made it possible for the employment of the 5 nm-thick Si_3_N_4_ and the HfO_2_ layer, which showed sufficient quality as the gate insulator. The device properties were electrically characterized and compared before and after proton irradiation. After proton irradiation, the direct current (DC) and RF performance were changed. The degree of the device performance alteration reduced when the irradiated proton energy increased. At low irradiated proton energy, the threshold voltage shift was larger in GaN-based MIS-HEMTs for the 5 nm-thick HfO_2_ gate insulator than in GaN-based MIS-HEMTs for the 5 nm-thick Si_3_N_4_ gate insulator, even though the HfO_2_ gate insulator exhibited stronger radiation resistance than the Si_3_N_4_ gate insulator. By contrast, the drain current and transconductance alteration were less when the HfO_2_ was applied for the gate insulator. Based on our previous study [20], using pulse-mode stress measurement and carrier mobility extraction, we revealed the device property alternation mechanisms, which were generated by proton irradiation. For the determination of the threshold voltage shift, the TID and DD effects were in competition, whereas the TID and DD effects were superposed for the drain current and transconductance reduction. The impact of proton irradiation on RF performance was also investigated.

## 2. Structure and Fabrication

We processed GaN-based MIS-HEMTs for the 5 nm-thick Si_3_N_4_ and HfO_2_ gate insulator on a 4-inch sapphire substrate. The epitaxial layers were grown by metal-organic chemical vapor deposition, which was composed of a 2 µm-thick GaN buffer, 50 nm-thick GaN channel, and 20 nm-thick AlGaN barrier layers. The Al content of the AlGaN barrier was 0.25. For the formation of Ohmic contact, Ti/Al/Ni/Au (30/100/30/100 nm) was deposited by an e-beam evaporation system followed by rapid thermal annealing at 850 °C for 40 sec. The phosphorus was implanted except for the active region of the GaN-based MIS-HEMTs, for the device isolation. To compare the radiation resistance and reveal the device property alternation mechanisms, two different gate dielectric layers (Sample-Si_3_N_4_: Si_3_N_4_ = 5 nm, Sample-HfO_2_: HfO_2_ = 5 nm) were prepared, which played as a surface passivation layer as well as the gate insulator. A Si_3_N_4_ layer was deposited by chemical vapor deposition (CVD), whereas a HfO_2_ layer was deposited by atomic layer deposition (ALD), respectively. The top of the Ohmic contact region and the source and drain contact pad areas were opened using a buffered oxide etch. For the gate electrode and contact pad formation, Ni/Au (30/370 nm) was deposited. The gate width (W_G_), gate length (L_G_), the distance between the source and gate (L_SG_), and the gate and drain (L_GD_) of the fabricated MIS-HEMTs were 100.0, 0.5, 1.0, and 3.5 µm, respectively. The radiated proton fluences were 10^15^ cm^−2^ at an energy of 5, 15, and 25 MeV, which were implemented at ARTI (Advanced Radiation Technology Institute). The flux of the irradiated proton was 1 × 10^12^ p/cm^2^·sec. The proton was irradiated at room temperature. The schematic cross-section of the processed GaN-based MIS-HEMTs for the 5 nm-thick Si_3_N_4_ and the HfO_2_ gate insulator was shown in Figure 1.

## 3. Results and Discussion

As shown in Figure 2, we measured the typical device transfer characteristics before and after proton irradiation in Sample-Si_3_N_4_ and Sample-HfO_2_. The drain current (I_D_) and gate leakage current (I_G_) were measured at drain bias (i.e., applied voltage at drain electrode, V_D_ = 4.5 V), while the gate bias (i.e., applied gate voltage at gate electrode, V_G_) was swept. The transconductance (g_m_) curves were achieved by the derivation of the I_D_. In the fresh device (i.e., before proton irradiation), there was a device property difference in threshold voltage (V_TH_), I_D_, and g_m_ for Sample-Si_3_N_4_ and Sample-HfO_2_, which resulted from the difference of the dielectric constant between Si_3_N_4_ and HfO_2_ [17].

The typical device transfer characteristics were changed by proton irradiation. For the two samples, the V_TH_ was positively shifted after proton irradiation at 5 MeV. The degree of the positive V_TH_ shift was larger in Sample-HfO_2_ than in Sample-Si_3_N_4_, despite the radiation resistance of HfO_2_ being better than that of Si_3_N_4_ [20]. Unlike the V_TH_ shift after proton irradiation at 5 MeV, the V_TH_ was negatively shifted after proton irradiation at 25 MeV. The superior radiation resistance of the HfO_2_ dielectric layer, which was compared to the Si_3_N_4_ dielectric layer, was confirmed by the lower I_G_ increase in Sample-HfO_2_ than in Sample-Si_3_N_4_. In contrast with the V_TH_ shift, the g_m_ and I_D_ reductions were less in Sample-HfO_2_ than in Sample-Si_3_N_4_.

For the quantization of the device property alteration and comparison of the immunity to the proton irradiation, several device parameters in terms of the threshold voltage shift (ΔV_TH_), drain current reduction (ΔI_D_), and transconductance maximum reduction (Δg_m,max_) were extracted according to the irradiated proton energy in Sample-Si_3_N_4_ and Sample-HfO_2_, as shown in Figure 3. After proton irradiation at 5 MeV, the threshold voltage was positively shifted. The ΔV_TH_ was reduced after proton irradiation energy at 15 MeV. When the irradiated proton energy was increased to 25 MeV, the V_TH_ was negatively shifted. In addition, the ΔV_TH_ was larger at 5 and 15 MeV in Sample-HfO_2_ than in Sample-Si_3_N_4_. However, the ΔI_D_ and Δg_m,max_ alteration tendency was different from the ΔV_TH_ tendency. The largest ΔI_D_ and Δg_m,max_ were obtained after proton irradiation at 5 MeV. The ΔI_D_ and Δg_m,max_ were gradually reduced when the irradiated proton energy became stronger. When we compared the Sample-Si_3_N_4_ and Sample-HfO_2_, the ΔI_D_ and Δg_m,max_ were less in Sample-HfO_2_. These results reflected that the device property alteration mechanism was not the same for ΔI_D_ and Δg_m,max_ and ΔV_TH_. 

The diminishment of the device property alteration with increases in the irradiated proton energy was due to the reduction of the linear energy transfer (LET), and/or non-ionizing energy loss (NIEL) [35,36], which was the energy loss rate through ionization and/or displacements. When the irradiated proton energy was increased, the irradiated proton speed increased and the LET and/or NIEL decreased. As a result, fewer ionized charges and defects in the dielectrics and semiconductors, respectively, were induced and the device property alteration was reduced.

To determine the 2DEG density variation, which was caused by proton irradiation, the capacitance-voltage C(V) measurements were carried out for the two samples before and after proton irradiation, as shown in Figure 4. The C(V) measurement frequency was fixed at 1 MHz while the gate bias was swept. After proton irradiation, the capacitance curves were positively and negatively shifted at low (5 MeV) and high (25 MeV) proton irradiation energy, respectively. The direction and degree of the capacitance curve shift corresponded to the ΔV_TH_. Therefore, the 2DEG density was varied by the proton irradiation, which resulted in the ΔV_TH_.

In Figure 5, the pulse-mode stress measurements were performed to achieve the key clue of the ΔV_TH_ mechanisms. When the pulses were applied to the GaN-based MIS-HEMTs, the electrical stress was induced and the charges were trapped inside the dielectric layer and/or semiconductor layers [17,20]. Therefore, the I_D_ was degraded by the reduction of the 2DEG density. When we conducted the pulse-mode stress measurements, the pulses applied at the drain electrode were increased from 0 V to 10 V, while the pulses at the gate electrode were fixed at V_TH_ + 2 V for the I_D_ measurements. The quiescent biases were set as (V_G_ = 0 V, V_D_ = 0 V), (V_G_ = V_TH_ − 2 V, 0 V), and (V_G_ = V_TH_ − 2 V, 10 V) for the without-stress, gate-stress, and gate-and-drain stress conditions, respectively. A 2 ms stress pulse and 0.2 µs measurement pulse were applied for inducing stress and measuring I_D_, respectively. Compared to the without-stress condition, the drain current was reduced under the gate-stress condition due to the trapped charges, which were mainly inside the gate insulator and AlGaN barrier [17,20,37]. Under the gate-and-drain stress condition, the I_D_ was lower than that of the gate-stress condition, since the impact of the traps existed in the GaN channel, and the GaN buffer was added to the gate-stress condition [17,20,37]. 

When the semiconductor was exposed to radiation, two different radiation effects (TID and DD effects) were generated. The TID effects mostly occurred in the interior of the dielectric layer [20,22,26] and were related to the LET. However, the TID effects generation mechanism was not the same for the dielectric layers. When the Si_3_N_4_ dielectric layer was deposited by the CVD system, there were many dangling bonds (so-called as *K* centers) [38,39,40]. When the Si_3_N_4_ was exposed to the radiation, the neutral *K*^0^ centers were converted to positively charged defects (K^+^ defects) [41]. On the other hand, the radiation exposure generated bond-breaking and oxygen vacancies in the HfO_2_ dielectric layer, which traps positive charges [42]. Despite the TID effects generation mechanism being different for the Si_3_N_4_ and HfO_2_ dielectric layers, the 2DEG density was increased and the V_TH_ was negatively shifted by the radiation in GaN-based MIS-HEMTs [20,27]. By contrast, inside the semiconductor, the DD effects, which were related to the NIEL, were introduced. The atomic displacement in the semiconductor was generated by the impinging energetic radiation, which resulted in lattice defects [43] in the AlGaN barrier, GaN channel, and GaN buffer layer. The electrons located at the hetero-interface were trapped at the lattice defects and the 2DEG density was reduced. As a result, the threshold voltage of the GaN-based MIS-HEMTs was positively shifted [44,45].

In Figure 5c, before proton irradiation, the ΔI_D,pulse_, which was the I_D_ reduction under gate-stress and gate-and-drain stress conduction, was slightly larger in Sample-Si_3_N_4_ than in Sample-HfO_2_. The reason for this was that more dangling bonds were included in the Si_3_N_4_ dielectric layer, which was in contrast to the HfO_2_ dielectric layer [20]. The ΔI_D,pulse_ gap, which was the ΔI_D_ difference between gate-stress and gate-and-drain stress conditions, was identical for the two samples, as the epitaxial layer was the same.

Through the pulse-mode stress measurements, the V_TH_ shift mechanism, which was generated by the proton irradiation, was explained. After proton irradiation the ΔI_D,pulse_ and ΔI_D,pulse_ gap was larger, which compared to the fresh devices due to the degradation of the dielectric layer and semiconductor quality. When we compared the two samples, the ΔI_D,pulse_ especially, under the gate-stress condition, was lower in Sample-HfO_2_ than in Sample-Si_3_N_4_. The HfO_2_ dielectric layer exhibited better immunity to proton irradiation, which compared to the Si_3_N_4_ dielectric layer. Therefore, in Sample-HfO_2_, the generation of the TID effects was less than in Sample-Si_3_N_4_. However, the ΔI_D,pulse_ gap stayed the same for the two samples at each proton irradiation energy. This phenomenon reflected the fact that the semiconductor quality degradation by the DD effects was the same for the two samples.

For the two samples, the same degree of positive V_TH_ shift was observed by the DD effects and the positive V_TH_ shift was compensated by the TID effects. In Sample-HfO_2_, less compensation occurred, which compared to the Sample-Si_3_N_4_, since the HfO_2_ dielectric layer showed superior radiation resistance to TID effects. Therefore, the ΔV_TH_ was rather larger in Sample-HfO_2_ than in Sample-Si_3_N_4_ even though the immunity to proton irradiation was stronger in the HfO_2_ dielectric layer than in the Si_3_N_4_ dielectric layer.

In Figure 6, the carrier mobility μ was extracted before and after the proton irradiation to determine the mechanisms of the Δg_m,max_ and ΔI_D_, since the Δg_m,max_ and ΔI_D_ depended on the μ. In the GaN-based MIS-HEMTs, there were parasitic resistance components, which resulted from the contact resistance and access region (between source and gate and gate and drain), and should be considered for the achievement of the precise μ behavior under the gated region. The μ behavior under the gated region, which took into account the parasitic components, was extracted, as is shown in Equation (1) [46]:μ = [(I_D_/V_D_Converted_)/L_G_]/(qW_G_N_S_)(1)
where V_D_Converted_ is the drain voltage across the gated region, q is the electron charge, and N_S_ is the electron concentration observed by the integration of the C(V) curve. The V_D_Converted_ was calculated as the following equation: V_D_Converted_ = V_D_ − I_D_(R_ACC_ − 2R_C_)(2)
where R_C_ and R_ACC_ are contact resistance and access region resistance, respectively, which are achieved using the transmission line method (TLM).

After the proton irradiation, the μ was degraded for the two samples. Consistent with the Δg_m,max_ and ΔI_D_, the largest μ degradation was observed at 5 MeV. When the irradiated proton energy was increased, the μ degradation was diminished for the two samples due to the reduced LET.

As shown in the inset in Figure 6a, the μ deterioration was observed from the low carrier density region after proton irradiation. The Coulomb scattering between the trapped charges and carriers located at the AlGaN/GaN hetero-interface was enhanced by the TID effects, since the amount of the trapped charges inside of the gate insulator was increased by the proton irradiation. The μ deterioration was also achieved at the high carrier density regime. The defects, which were induced by the proton irradiation and located near the AlGaN/GaN hetero-interface, were other μ degradation components. In addition, the overall tendency of the μ maximum reduction was the same as for the Δg_m,max_ and ΔI_D_, as shown in Figure 6b. Taken together, the origin of the Δg_m,max_ and ΔI_D_ was due to the μ degradation, which was determined by the superposition of the TID and DD effects in the GaN-based MIS-HEMTs.

Before and after proton irradiation, the sheet resistance of Sample-HfO_2_ and Sample-Si_3_N_4_ were extracted by transmission line method (TLM), as seen in Table 1. The sheet resistance increased after proton irradiation because of the TID and DD effects. With increasing irradiated proton irradiation energy, the increase of the sheet resistance was reduced. In Sample-HfO_2_, the sheet resistance increase was less, which compared to Sample-Si_3_N_4_.

Note that the contact resistance was also extracted through the TLM before and after proton irradiation for the two samples. As shown in Table 2, it was difficult to observe the contact resistance alteration resulting from the proton irradiation. Therefore, the source and drain electrodes were unaffected by the proton irradiation for the two samples.

In Figure 7, the RF characteristics were investigated before and after proton irradiation in Sample-HfO_2_ and Sample-Si_3_N_4_. For the extraction of the cut-off frequency f_T_ and maximum oscillation frequency f_MAX_, we measured the S-parameter using the network analyzer. First, the measured S-parameter was converted to the H-parameter. We fitted a linear line with a −20 dB slope on the H21 curve for achieving the f_T_. The f_T_ was determined by a point, which was the extrapolated point of the linear line to the 0 dB. For the extraction of f_MAX_, we converted the measured S-parameter to the maximum stable gain MSG/maximum available gain MAG. A linear line with a −20 dB slope was then fitted at stability factor K = 1. The extrapolated point of the linear line to the 0 dB was defined as the f_MAX_.

The RF performance in terms of f_T_ and f_MAX_ was also deteriorated by the proton irradiation for the two samples. When the proton was irradiated at 5 MeV, f_T_ and f_MAX_ were degraded by 30% and 45%, respectively, in Sample-Si_3_N_4_. Compared to Sample-Si_3_N_4_, the RF performance degradation was less in Sample-HfO_2_. Like with the DC parameters, the RF performance deterioration was diminished when the irradiated proton energy was stronger. When we compared the RF performance degradation to the DC performance degradation, the RF performance degradation was much larger, as the defects and traps generated by the radiation were more sensitive to the frequency response [47].

## 4. Conclusions

The mechanisms of the device property alteration, which were generated by proton irradiation, were studied through the electrical device characterization in GaN-based MIS-HEMTs. The device properties in terms of the threshold voltage, drain current, and transconductance varied by the proton irradiation due to the generation of the total ionizing dose (TID) and displacement damage (DD) effects. We conducted the capacitance measurements, pulse-mode stress measurements, and carrier mobility extraction, and revealed the origin of the device property alteration. The threshold voltage shift was determined by the competition between the TID and DD effects. However, drain current and transconductance degradation resulted from the superposition of the two different radiation effects. The device property alteration was diminished when the irradiated proton energy became stronger due to the reduction of the energy loss rate. The deterioration of the RF performance, such as cut-off and maximum oscillation frequency, was large, since the defects and traps generated by proton irradiation were sensitive to the frequency response.

## Figures and Tables

**Figure 1 nanomaterials-13-00898-f001:**
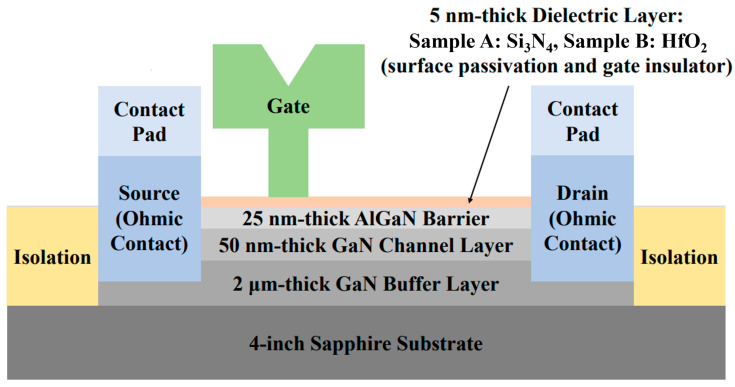
Schematic cross-section of the processed GaN-based MIS-HEMTs for the 5 nm-thick Si_3_N_4_ and HfO_2_ gate insulator. The 5 nm-thick Si_3_N_4_ and the HfO_2_ layer were employed for Sample-Si_3_N_4_ and Sample-HfO_2_, respectively, which acted as the gate insulator and surface passivation.

**Figure 2 nanomaterials-13-00898-f002:**
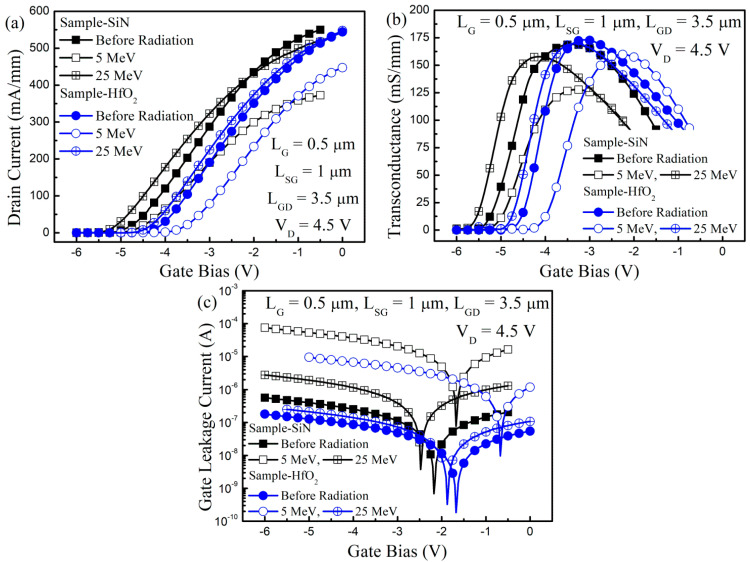
Typical transfer characteristics comparison of GaN-based MIS-HEMTs for the 5 nm-thick Si_3_N_4_ and HfO_2_ gate insulator before and after proton irradiation at 5 and 25 MeV. (**a**) Drain current; (**b**) transconductance; (**c**) gate leakage current versus gate bias.

**Figure 3 nanomaterials-13-00898-f003:**
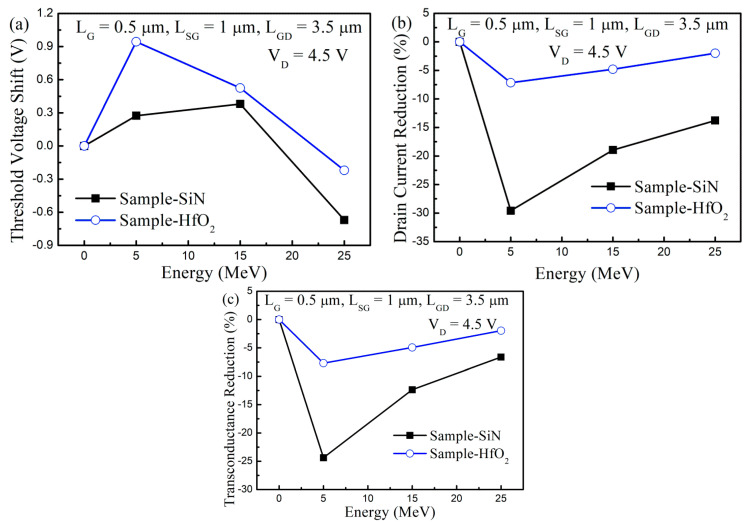
Device parameter extraction before and after proton irradiation in GaN-based MIS-HEMTs for the 5 nm-thick Si_3_N_4_ and HfO_2_ gate insulator. (**a**) Threshold voltage shift; (**b**) transconductance maximum reduction; (**c**) drain current reduction as a function of the proton irradiation energy.

**Figure 4 nanomaterials-13-00898-f004:**
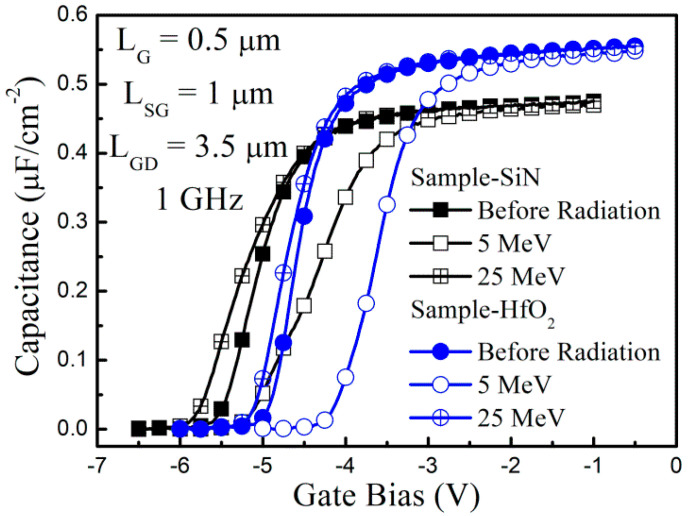
Capacitance-voltage measurement results before and after proton irradiation in GaN-based MIS-HEMTs for the 5 nm-thick Si_3_N_4_ and the HfO_2_ gate insulator.

**Figure 5 nanomaterials-13-00898-f005:**
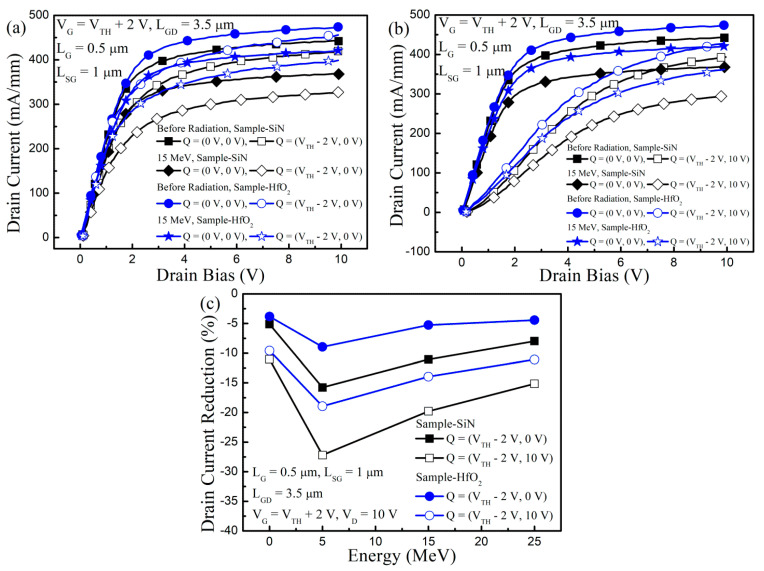
Pulse-mode stress measurement results in GaN-based MIS-HEMTs for the 5 nm-thick Si_3_N_4_ and the HfO_2_ gate insulator. Drain current versus drain bias with the quiescent bias of: (**a**) (0 V, 0 V) and (V_TH_ − 2 V, 0 V); (**b**) (0 V, 0 V) and (V_TH_ − 2 V, 10 V) before and after proton irradiation at the energy at 15 MeV; (**c**) drain current reduction from the quiescent bias of (0 V, 0 V) induced by the pulse-mode stress with the quiescent bias of (V_TH_ − 2 V, 0 V) and (V_TH_ − 2 V, 0 V) as a function of irradiated proton energy.

**Figure 6 nanomaterials-13-00898-f006:**
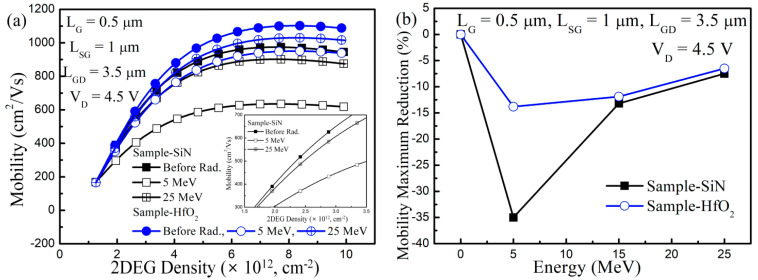
Carrier mobility extraction before and after proton irradiation in the GaN-based MIS-HEMTs for the 5 nm-thick Si_3_N_4_ and the HfO_2_ gate insulator. (**a**) Carrier mobility vs 2DEG density before and after proton irradiation at 5 MeV and 25 MeV; (**b**) Carrier mobility maximum reduction vs. the irradiated proton energy. Inset in (**a**) shows the zoomed-in carrier mobility behavior at a low 2DEG density regime in GaN-based MIS-HEMTs for the 5 nm-thick Si_3_N_4_ gate insulator before and after proton irradiation at 5 MeV and 25 MeV.

**Figure 7 nanomaterials-13-00898-f007:**
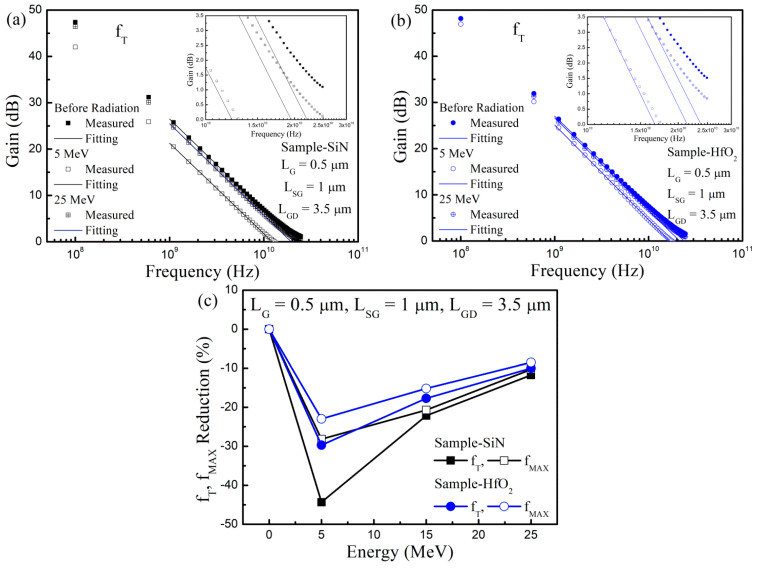
Radio-frequency performance investigation, such as cut-off and maximum oscillation frequency before and after proton irradiation in the GaN-based MIS-HEMTs for the 5 nm-thick Si_3_N_4_ and HfO_2_ gate insulator. Cut-off frequency characteristics before and after proton irradiation in GaN-based MIS-HEMTs for a 5 nm-thick (**a**) Si_3_N_4_ and (**b**) HfO_2_ gate insulator. (**c**) Extracted cut-off frequency and maximum oscillation frequency as a function of the irradiated proton energy.

**Table 1 nanomaterials-13-00898-t001:** Extracted sheet resistance before and after proton irradiation in GaN-based MIS-HEMTs for the 5 nm-thick HfO_2_ and the Si_3_N_4_ gate insulator.

	Before Proton Irradiation (Ω/χ)	Proton Irradiation at 5 MeV (Ω/χ)	Proton Irradiation at 15 MeV (Ω/χ)	Proton Irradiation at 25 MeV (Ω/χ)
Sample-HfO_2_	453	562	518	491
Sample-Si_3_N_4_	451	522	499	482

**Table 2 nanomaterials-13-00898-t002:** Extracted contact resistance before and after proton irradiation in GaN-based MIS-HEMTs for the 5 nm-thick HfO_2_ and the Si_3_N_4_ gate insulator.

	Before Proton Irradiation (Ω/mm)	Proton Irradiation at 5 MeV (Ω/mm)	Proton Irradiation at 15 MeV (Ω/mm)	Proton Irradiation at 25 MeV (Ω/mm)
Sample-HfO_2_	1.24	1.26	1.23	1.25
Sample-Si_3_N_4_	1.25	1.24	1.24	1.25

## Data Availability

Not applicable.

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
