# Peer review of "Mechanisms of the Device Property Alteration Generated by the Proton Irradiation in GaN-Based MIS-HEMTs Using Extremely Thin Gate Insulator"

_nanomaterials, 2023, doi:10.3390/nano13050898_

Round 1

Reviewer 1 Report

This study investigates the mechanisms behind the changes in transistor parameters, such as threshold voltage, transconductance, maximum current, and voltage gain, after exposure to gamma radiation. The findings of this research have important implications for aerospace applications, such as satellite RF communication links, and are suitable for publication in the scientific community.

The study was well-conducted and the results are robust and effectively presented in a clear and concise manuscript. However, to provide a complete understanding of the study's outcomes and significance, it would be valuable to discuss the trade-offs involved in the minimum required dielectric layer thickness for radiation resistance, including the relationship between the thickness of the dielectric layer and transistor bandwidth.

It is noted that the study did not specifically address dielectric layer thickness as a variable, and I am not asking to include this study. Nonetheless, given that the title of the manuscript highlights the use of ultra-thin dielectric layers, referred to as "extremely thin gate insulators," with a thickness of only 5 nm, it would be appropriate to include some comments about this aspect in the concluding remarks of the manuscript.

Reviewer 2 Report

Referee report on “Mechanisms of the Device Property Alteration Generated by  the Proton Irradiation in GaN-based MIS-HEMTs using Extremely Thin Gate Insulator

This is a very good, technically sound and application oriented article that can and should certainly be recommended for publication.

1.     However, I am sure that one cannot talk about the mechanism and at the same time evade the discussion about the point defect production in the conditions of displacement damage, that is, in conditions when new vacancies and interstitials are created. Obviously, more general information about these processes should be added to the Introduction. In addition, information about the hierarchies of the corresponding displacement energies would be useful.

Patrick, Erin, et al. "Modeling proton irradiation in AlGaN/GaN HEMTs: Understanding the increase of critical voltage." IEEE Transactions on Nuclear Science 60.6 (2013): 4103-4108.

Recent information on point defect in other materials, shown in Fig. 1, can be found (and references therein): Ananchenko, D. V., et al. "Radiation-induced defects in sapphire single crystals irradiated by a pulsed ion beam." Nuclear Instruments and Methods in Physics Research Section B: Beam Interactions with Materials and Atoms 466 (2020): 1-7.   Feldbach, Eduard, et al. "Defects induced by He+ irradiation in γ-Si3N4." Journal of Luminescence 237 (2021): 118132.

This is important to attract more reader interest and further incentive applications.

2.     Line 64. It is not clear why the correct chemical formula is used for all materials except for Si3N4?  The same question is also for Fig.1.

3.     Line 134. Can you clearly state what you mean by the word “radiation hardness”?

4.     Could you briefly give information about what TRIM calculations predict for proton exposures low (5 MeV) and high (25 MeV).

Therefore, in the conclusions, it is necessary to clearly formulate what new data about the studied materials were obtained in this work?  

In general, the manuscript is interesting and can be considered for publication after constructive reflection on the above comments.

Reviewer 3 Report

This paper studies proton irradiation effects on the transistor device property change. More details of the experiments are needed, and microstructural characterization effort is needed. See the comments below:

1. For the proton irradiation experiments, can authors show the dpa (displacement per atom) w.r.t the dose using SRIM software? Also, what irradiation temperature was selected to use, and is it related to the space application? What’s the dose rate used for proton irradiation?

2. In addition, I don’t see a detailed discussion of the effects of irradiation-induced point defect (vacancy, interstitial) and extended defects (such as dislocations) on the performance of the transistor?

3. I think microstructural characterization is needed to show the irradiation induced defects, which results in the property change of the transistor.

Reviewer 4 Report

General comments:

The fact of negative charges trapped in the dielectric is mentioned in the document. Since electrons generally escape under an electric field, I wonder if negative trapping can be thought of as a sort of "floating gate" where charge behaves as if it is stored in a parasitic capacitor and can only be removed by tunneling effects . Please comment.

In general, articles that study the effects of radiation also present methods for sustaining or recovering devices after irradiation. Hot annealing processes are among the most effective for removing trapped charges. There are no hints of annealing in the document while these would really complement the performance of the device. Can the authors add some comments?

Some point comments:
- line 68: rephrase the sentence "there are two different radiation effects are induced."
- line 69: replace effects with effect in "effects and another is displacement damage (DD) effects."
- lines 74-75: correct the plural in " To reveal the device property alteration mechanisms, which is caused by the proton irradiation, "
- line 92: again, " which was generated" or " which were generated" ?
- line  322: rephrase "When the proton was irradiated, "

Round 2

Reviewer 2 Report

After a sufficiently constructive revision, the manuscript can be recommended for publication.

Reviewer 3 Report

I have no more comments.

Reviewer 4 Report

As a second review of the paper, I find all the previous comments being addressed. I only wish to come back again on the question of the negative trapped charges.

On line 42 the negative trapped charges at the barrier surface are mentioned even I imagine these have nothing to do with the carrieds trapped in the substrate. Can you confirm or explain little better?

On line 219 you mention "The electrons located at the hetero-interface are trapped " so, are not these negative trapped charges ? please comment
